# A Similar Lifetime CV Risk and a Similar Cardiometabolic Profile in the Moderate and High Cardiovascular Risk Populations: A Population-Based Study

**DOI:** 10.3390/jcm10081584

**Published:** 2021-04-09

**Authors:** Małgorzata Chlabicz, Jacek Jamiołkowski, Wojciech Łaguna, Paweł Sowa, Marlena Paniczko, Magda Łapińska, Małgorzata Szpakowicz, Natalia Drobek, Andrzej Raczkowski, Karol A. Kamiński

**Affiliations:** 1Department of Population Medicine and Lifestyle Diseases Prevention, Medical University of Białystok, 15-259 Białystok, Poland; mchlabicz@op.pl (M.C.); jacek909@wp.pl (J.J.); mailtosowa@gmail.com (P.S.); m.paniczko@gmail.com (M.P.); magda.lapinska@umb.edu.pl (M.Ł.); malgorzata.szpakowicz@umb.edu.pl (M.S.); drobeknatalia@gmail.com (N.D.); andrzej.raczkowski@umb.edu.pl (A.R.); 2Department of Invasive Cardiology, Teaching University Hospital of Białystok, 15-259 Białystok, Poland; 3Faculty of Computer Science, Bialystok University of Technology, 15-259 Białystok, Poland; wojciech.laguna@gmail.com; 4Department of Cardiology, Teaching University Hospital of Białystok, 15-259 Białystok, Poland

**Keywords:** cardiometabolic profile, cardiovascular risk, population studies

## Abstract

Background: Cardiovascular disease (CVD) is a major, worldwide problem that remains the dominant cause of premature mortality in the world, and increasing rates of dysglycaemia are a major contributor to its development. The aim of this study was to investigate the cardiometabolic profile among patients in particular cardiovascular risk classes, and to estimate their long term CV risk. Methods: A total of 931 individuals aged 20–79 were included. The study population was divided into CV risk classes according to the latest European Society of Cardiology recommendations. Results: Most of the analyzed anthropometric, body composition and laboratory parameters did not differ between the moderate and high CV risk participants. Interestingly, estimating the lifetime risk of myocardial infarction, stroke or CV death, using the LIFEtime-perspective model for individualizing CardioVascular Disease prevention strategies in apparently healthy people, yielded similar results in moderate and high CV risk classes. Conclusion: The participants who belonged to moderate and high CV risk classes had very similar unfavorable cardiometabolic profiles, which may result in similar lifetime CV risk. This may imply the need for more aggressive pharmacological and non-pharmacological management of CV risk factors in the moderate CV risk population, who are often unaware of their situation. New prospective population studies are necessary to establish the true cardiovascular risk profiles in a changing society.

## 1. Introduction

Cardiovascular disease (CVD) is highly widespread and remains the dominant cause of premature mortality in the world [1]. Cardiometabolic disorders, i.e., obesity, insulin resistance, diabetes, hypertension and dyslipidaemia are known CVD risk factors, which contribute to the prevalence of CVD [2,3]. Recently, a considerable increase was seen in the prevalence of obesity, metabolic syndrome and diabetes mellitus worldwide [4,5], while the prevalence of hypertension was stable [6]. Many studies verify that healthy lifestyles and the control of cardiometabolic disorder prevent CVD events [7]. Nevertheless, many of these disorders can remain undiagnosed, and thus, not treated [8]. Consequently, special attention should be paid to people with undiagnosed CVD and with increased CV risk, who consider themselves healthy and may be reluctant to subject themselves to diagnostics and therapy. There is little data on the cardiometabolic profile of individuals with moderate CV risk, who generally consider themselves healthy.

Cardiovascular (CV) risk means the probability of a CVD event of atherosclerotic origin in an individual within a specified period of time. The level of risk depends on the presence of many individual characteristics and environmental factors, which are known as risk factors [9]. Recent guidelines recommend the use of lifetime risk-assessment in connection with short-term estimation. Therefore, in addition to short-term CV risk assessment, in the form of the Systematic Coronary Risk Estimation system and Framingham Risk Score, we used LIFEtime-perspective model for individualizing CardioVascular Disease prevention strategies in apparently healthy people (LIFE-CVD) to assess lifetime CV risk [10].

In our clinical work, we noticed a reluctance to take preventive actions in the moderate cardiovascular risk class, not only among patients, but also in the medical community. We understand from clinical experience that these people often consider themselves healthy. They disregard recommendations and such attitudes are even supported by some doctors. The general aim of this study was to investigate whether people with moderate cardiovascular risk (who often consider themselves healthy), are clinically closer to the high-risk class than the low risk one.

The present study aims to: (1) Describe the general population according to CV risk classes using the most recent 2019 ESC/EAS guidelines for the management of dyslipidaemias [11], also in terms of their risk assessed using other risk scores; and (2) compare clinical profiles (laboratory tests, echocardiographic parameters, anthropometric measurements, body composition analysis and subjective well-being), with particular attention to cardiometabolic profile, between individuals in different CV risk classes.

## 2. Methods

### 2.1. Data Collection

The data were collected through standardized health examinations in specially equipped examination center. The details of the subjects’ medical history were collected from questionnaires at the time of study entry: Demographic data, CV risk factors and the history of cardiovascular events. A comprehensive assessment was performed.

Anthropometric measurements included measurement of height, weight, waist, abdomen, hips and thigh. Measurements were performed in accordance with the World Health Organization (WHO) guidelines [12]. The subject was standing upright during the measurements, with their arms relaxed by their side, feet evenly spread apart and body weight evenly distributed. The measurements were taken with the SECA 201 tape (SECA, Hamburg, Germany). The waist circumference was measured horizontally by a metal tape in the mid-axillary line midway between the lowest rim of the rib cage and the tip of the hip bone. The hip circumference was measured around the widest circumference of the buttocks, with the tape parallel to the floor. The thigh circumference was measured directly below the gluteal fold. Both thighs were measured, and the mean value was calculated as the final thigh circumference. The waist to hip ratio (WHR) was calculated as a ratio between the waist and hips circumference.

Body mass index (BMI) was calculated as weight in kilograms divided by height in meters squared.The blood pressure (BP) was measured while the participants were seated after a minimum rest period of 5 min using the oscillometeric method applying Omron Healthcare Co., Ltd. MG Comfort (HEALTHCARE Co., Ltd. Terado-cho, Muko, Kyoto, Japan) [8].

In echocardiography (ECHO) the dimensions of the heart, left atrial (LA) volume, left ventricular ejection fraction (LVEF) were taken by using the biplane analysis (Vivid 9, GE Healthcare, Chicago, IL, USA). The LVH was defined as LVMI ≥ 115 g/m^2^ for men and ≥95 g/m^2^ for women; and left atrial volume index (LAVI) was calculated, with enlarged volume defined as >34 mL/m^2^. Diastolic dysfunction of left ventricle was assessed based on the latest recommendations [13].

The carotid ultrasound was performed using the two-dimensional ultrasound (Vivid 9, GE Healthcare, Chicago, IL, USA). The left and right common carotid artery (CCA), carotid bifurcations, internal (ICA) and external (ECA) carotid artery were examined for wall thickness and the presence of atherosclerotic plaque [14]. The severity of stenosis was determined using the North American Symptomatic Carotid Endarterectomy Trial (NASCET) criteria [15].

The body composition was measured by dual energy x-ray absorptiometry (DEXA) (GE Healthcare, Chicago, IL, USA) as described previously [16]. The A/G ratio was calculated between the fat of the android (central) and fat of the gynoid (hip and thigh) regions. Fat mass index (FMI) was calculated as fat in kilograms divided by height in meters squared. 

### 2.2. Biochemical Parameters

Peripheral intravenous fasting blood samples were collected at the time of visit in the morning after eight h of fasting. Next, the samples were immediately centrifuged and stored at −70 °C until analysis. Total cholesterol (TC), triglycerides (TG), and creatinine were determined by the enzymatic colorimetric method on the Cobas c111 from ROCHE. The high-density lipoprotein cholesterol (HDL-C) and low-density lipoprotein cholesterol (LDL-C) were determined by the homogeneous enzymatic colorimetric method on the Cobas c111 from ROCHE (ROCHE, Meylan, Isère, France). Fasting glucose and the 120 min glucose in oral glucose tolerance test (OGTT) were measured by the enzymatic reference method with hexokinase on the Cobas c111 from ROCHE. Fasting insulin, the 120 min insulin in OGTT, fasting C-peptide, and the 120 min C-peptide in OGTT were measured by the electrochemiluminescence method ECLIA on the Cobas e411 from ROCHE. Hemoglobin A1c (HbA1c) was determined by ion-exchange high performance liquid chromatography (HPLC) on D-10 from Bio-Rad (Bio-Rad, Hercules, CA, USA). N-terminal pro-brain natriuretic peptide (NT-proBNP) and high-sensitivity troponin T (hs-TnT) were determined by the electrochemiluminescence method on the Cobas e411 from ROCHE. Homeostatic model assessment for insulin resistance (HOMA-IR) was calculated from the following formula: score = fasting insulin (µU/mL) × fasting glucose (mmol/L)/22.5 [17].

### 2.3. Cardiovascular Risk Assessment

The study population was divided into CV risk classes according to the latest recommendation: “2019 ESC/EAS guidelines for the management of dyslipidaemias: lipid modification to reduce cardiovascular risk” [11]. Firstly, high and very-high risk individuals were identified. Due to the lack of a clear definition in the guidelines of “significant plaque” on carotid ultrasound, we extrapolated guidelines of coronary computed tomography (CT) scan. Probands with >50% ICA stenosis were qualified to the very-high CV risk class. Next the Systematic Coronary Risk Estimation (SCORE) was calculated. It was recalibrated in Poland [18], thus, we used Pol-SCORE system to assess the 10-year risk of fatal CV based on the following risk factors: Age, gender, smoking, BPs, and total cholesterol for individuals aged 40–70 [18,19]. Pol-SCORE was calculated for participants who were not pre-qualified to the high and very-high CV risk classes according the above-mentioned recommendations. A Pol-SCORE <1% was defined as a low risk, a Pol-SCORE from 1% do <5% as a moderate risk, a Pol-SCORE from 5% to <10% as a high risk, and a Pol-SCORE ≥10% as a very-high risk. Finally, based on information about atherosclerotic cardiovascular disease (ASCVD), diabetes mellitus (DM), estimated glomerular filtration rate (GFR), prevalence of CVD risk factors, and the Pol-SCORE value, the total CV risk class for each participant was established [11].

Framingham Risk Score (FRS) predicting a 10-year risk of developing the first CVD event (coronary death, MI, coronary insufficiency, angina, ischemic stroke, hemorrhagic stroke, transient ischemic attack, peripheral artery disease, or heart failure) using scores for lipids or BMI based on the following factors: age, diabetes, smoking, treated and untreated BPs, total cholesterol, HDL-c or BMI replacing lipids [20]. This 2018 calculator has been provided by the American Heart association and the American College of Cardiology.

The LIFEtime-perspective model for individualizing CardioVascular Disease prevention strategies in apparently healthy people (LIFE-CVD) estimates the probability of survival free of heart attack or stroke, a 10-year risk of MI, stroke or CV-death, and a lifetime risk of MI, stroke or CV-death using the following factors: Age, gender, smoking, geographic region, diabetes, parental history of MI prior to age 60, BPs, BMI, total cholesterol, HDL-c and LDL-c [21].

### 2.4. Subjective Well-Being Assessment

The subjective well-being was measured by the Satisfaction with Life Scale (SWLS) developed by Diener [22]. The Euro Quality of Life Visual Analogue Scale (EQ-5D) was designed to measure health related quality of life. The Visual Analogue Scale (EQ-VAS) is the second part of the above questionnaire, asking patients to mark health status on the day of the interview [23]. Depression symptoms were assessed by the Beck Depression Inventory (BDI) [24], a self-report measurement to assess severity of depression.

### 2.5. Trial Registration and Ethical Issues

ClinicalTrials.gov Identifier: NCT03197363. Ethical approval for this study was provided by the local Ethics Committee (approval number: R-I-002/108/2016).

### 2.6. Statistical Analysis

Descriptive statistics for quantitative variables were presented as means and standard deviations and as counts and frequencies for qualitative variables. Comparisons of variables between subgroups were conducted using the Dwass-Steele-Critchlow-Fligner test [25]. Regression models were presented using regression coefficients, *p*-value of Wald tests and coefficients of determination for the model (R2). Statistical hypotheses were verified at a 0.05 significance level. The IBM SPSS Statistics 20.0 statistical software (Armonk, NY, USA) was used for all calculations.

## 3. Results

The study was conducted in 2017–2020 on a representative sample of area residents. Randomly selected residents (2449) from the mayor’s office database were invited to participate in the study. A total of 966 individuals responded and were examined. Due to incomplete data, 35 people were excluded from further analysis. As a result, 931 individuals were included in the research group out of which 63 (6.8%) participants had established CVD, e.g., previous myocardial infarction (MI) 21 (2.3%), stable coronary heart disease (CHD) 18 (1.9%), peripheral artery disease (PAD) 8 (0.9%), stroke 16 (1.7%). Moreover, 13 (1.4%) participants had a carotid plaque >50% in internal carotid artery, 275 (29.6%) history of hypertension (AH), 71 (7.6%) history of diabetes, 27 (2.9%) history of atrial fibrillation (AF) and 186 (20.1%) currently smoked cigarettes. The baseline characteristics of the study population are summarized in Table 1.

The mean age was 49.1 [Interquartile range (IQR) 20–79] years and 43.2% male. The mean Pol-SCORE risk was 4.0 ± 4.9%, the mean BMI was 26.8 ± 5.0 and 20.1% was currently smoking. Percentages of the low-risk, moderate-risk, high-risk and very-high CV risk classes were 46.1%, 22.8%, 13.5%, 17.6%, respectively. Women were more often at the low-risk class and less often at the very-high risk class compared to men (*p* < 0.001). Individuals in higher CV risk classes were older, more likely to be male, had higher systolic (BPs) and diastolic blood pressure (BPd), more frequently declared history of hypertension (AH), diabetes mellitus (DM), atrial fibrillation (AF). However, there was no difference regarding smoking (*p* = 0.660).

In the laboratory study evaluation, most parameters did not differ between the moderate and high CV risk classes. Whereas, they differed statistically between the low and moderate CV risk classes, i.e., NT-proBNP, fasting glucose, the 120-min glucose in OGTT, fasting insulin, the 120-min insulin in OGTT, fasting C-peptide, the 120-min C-peptide in OGTT, LDL-C, TG, hs-CRP. The same phenomenon was observed in anthropometric measurements and body composition analysis—in the moderate and high CV risk groups an abdominal distribution of adipose tissue was observed. There was no significant difference between the moderate and high-risk class in the following parameters: BMI, waist circumference, hips circumference, abdominal obesity (waist >80 cm in women, >94 cm in men), WHR, FMI, total fat mass, android fat, visceral mass and A/G fat mass. The BPs, BPd, percentage of elevated BP ≥ 140 and/or ≥90 mmHg, left ventricular mass index (LVMI), LVH and left atrial volume index (LAVI) present the same relationship. The detailed characteristics of the subgroup study population are presented in Table 1 and Table 2. Figure 1 shows schematic presentation of the relative differences in selected anthropometric, body composition and laboratory parameters between CV risk classes.

An interesting result was the lack of differences between all CV classes in terms of muscle mass: total lean mass, android lean mass, gynoid lean mass and legs lean mass did not differ statistically between all classes. The details are presented in Table 2.

Another unexpected finding of the study is the subjective well-being analysis. Measured by SWLS, well-being was not related to the level of CV risk, also between the moderate and high CV risk groups, as opposed to health related quality of life and depression. The EQ-5D value decreased in the higher risk class while BDI value increased, but there was no significant difference between the moderate and high-risk CV classes. The details are presented in Table 2.

In Table 3, we present the medical history of the study population in the context of CV risk, with particular emphasis on well-controlled CV risk factors and undiagnosed diseases. The moderate and high-risk CV classes did not differ significantly in the percentage of individuals with a history of hypertension, undiagnosed hypertension, a history of hypercholesterolaemia, undiagnosed hypercholesterolaemia and a history of diabetes.

The estimated CV risks using short-term tools differed significantly between the classes, but not the estimated current lifetime risk of MI, stroke of CV death using LIFE-CVD calculator. There was no significant difference between the moderate and high CV risk participants regarding the estimated lifetime CV risk. The second conclusion that should be drawn from this analysis is a very high probability of a fatal CV 10-year risk calculated using Pol-SCORE comparing to a 10-year risk of MI, stroke or CV-death using Life-CVD in the groups of high and very-high CV risk. The probability of a fatal CV 10-year risk is higher than a 10-year risk of MI, stroke or CV-death, which may suggest overestimation of a CV-death risk using the Pol-SCORE scale or underestimation with LIFE-CVD in these CV classes. Details are presented in Table 4.

## 4. Discussion

The present study gives an insight into the clinical characteristics of the CV risk classes stratification in general population. The participants belonging to the moderate and high CV risk classes have very similar, unfavourable cardiometabolic profiles, which may result in a similar lifetime CV risk.

CVD is still the leading cause of death and disability in the European population [26], as well as in Poland [27,28,29]. It is often silent and may occur suddenly, underscoring the importance of prevention. It is vital that individuals at risk of developing CVD can be effectively identified and properly treated according to their risk. The patients with risk factors that are controlled could have a lower probability of developing complications than patients whose CV risk factors are unrecognized or poorly controlled [30]. The treatment of hypertension reduced the risk of CHD by approximately 25% [31], and the addition of lipid-lowering therapy in patients with hypertension reduced the residual risk of CHD by more than 35% [32]. Niklas et al. [33] showed that female sex, no smoking, comorbid CVD or diabetes and the frequency of medical visits correlated with better control of hypertension and hypercholesterolemia. This suggests that patients at high risk are aware of their health and use medical care more often. In Poland, over 60% of people with hypercholesterolemia and about 40% with hypertension are not aware of their condition [34]. Therefore, it is very important to identify patients with cardiovascular risk factors, especially those in the moderate cardiovascular risk classes, who may not be aware of their increased risk and may not take action to minimize or reduce it.

### 4.1. Cardiovascular Risk Categories

In the current study, less than half of the population were in the low-risk class. Women were more often in the low-risk class and less often in the very-high risk class compared to men. Amor et al. [35] showed that, in Spain, most men were at a moderate risk, while women were at low CV risk—the participants were aged between 40 to 65. Polak et al. [36] showed CV risk categories of participants in the Polish edition of the Health, Alcohol, and Psychosocial factors in Eastern Europe (HAPIEE) Project, in which participants were aged 45–69 years. According to the above, 28% of men and 60% of women were in the low or moderate CV risk category, 23.1% of men and 10.5% of women were in the high category, and 48.2% of men and 29.6% of women were in the very-high CV risk category. In this study, the SCORE risk algorithm developed in 2003 was used, thus, it could overestimate the 10-year mortality rate. Diederichs et al. [37] estimated a 10-year risk of fatal CVD in the general population of 40 to 69-year-olds without a history of CVD in Germany. The prevalence of low, moderate and high risk was 42.8%, 38.5% and 18.8% in men and 73.7%, 18.1% and 8.2% in women. However, women are more often in the low-risk category, and less often in the very-high risk class compared to men, which has been confirmed by our study.

### 4.2. Comparison of Parameters between CV Groups

Excessive body weight is a major public health problem, and two thirds of the adult population in the United States, and at least half of the population in many developed countries are overweight and obese. BMI is an accepted parameter to define the level of general obesity and is correlated with CVD [5]. In Poland, 23.6% of men and 19.7% of women are obese [38]. In contrast, our study found that the prevalence of overweight and obese individuals was 37.8%, and 26.7%, respectively. Abdominal obesity has been defined as the fat located around the viscera and within peritoneum, and is identified as a predictor of adverse metabolic or CV outcomes independently of BMI [39,40]. Furthermore, measures of insulin resistance correlate significantly with intra-abdominal adiposity in humans [41]. Fan et al. [42] revealed that general overweight and obesity was not significantly associated with the risk of atherosclerotic cardiovascular disease, but it was connected with a higher level of fasting glucose, TG, LDL-C, uric acid, hs-CRP, increased BPs and lower education level in elderly Chinese subjects. While, abdominal obesity was significantly associated with older age, an increased risk of atherosclerotic cardiovascular disease, higher levels of fasting glucose, TG, LDL-C, uric acid, hs-CRP, increased BPs and lower education level. Diabetes mellitus is a growing problem both, in the general population and among patients with CVD [43], and increased glycaemic levels are positively correlated with vascular complication [44,45,46]. Moreover, there is evidence that impaired glucose tolerance increases CV risk. On the other hand, abdominal obesity may play a major role in insulin resistance and various metabolic risk factors [47]. In the current study, half of the population had incorrect WHR (≥0.85 women, ≥0.9 men) and 68.8% participants had abnormal abdominal circumference (>80 cm women, >94 cm men). The participants who belonged to the moderate and high CV risk classes had a similar unfavourable cardiometabolic and body composition profile, which is most likely associated with abdominal fat distribution.

In Poland, a small percentage of CVD is optimally treated [48], and a large number of people have undiagnosed risk factors. In the WOBASZ study, the prevalence of hypertension was 42.7%, awareness 59.3%, and control 23% [32]. In the Polish NATPOL study, in general population aged 18–79, the prevalence of hypercholesterolemia was 61.1%, and the efficacy of treatment at 10.9% (TC < 4.9 mmol/L) [49]. In POLASPIRE survey, 42% of the participants with established CAD had uncontrolled hypertension, 62% uncontrolled hypercholesterolemia, 22% high fasting glucose, 17% were smokers and 42% were obese [48]. Our results are in accordance with the findings from other Polish studies.

The current study displayed no relationship between SWLS and CV risk classification. The quality-of-life assessment has become more popular due to its impact on chronic diseases. In line with our outcomes, Valkamo et al. [50] suggested that life satisfaction is not determined by the severity of CVD, while Steca ey al. [51] showed that life satisfaction measured by SWLS was negatively correlated with the diagnosis and illness perception among patients with CVD undergoing a rehabilitation program. Therefore, we have conducted an additional analysis (Appendix A), and we revealed that life satisfaction was not related to the actual risk factors, such as the presence of hypertension, hypercholesterolaemia or hyperglycemia, while a strong unfavorable relationship was shown with the diagnosis of the disease itself. This may lead to the conclusion that participants do not want to know about diseases that affect CV risk, because it could lower their life satisfaction. This may be an underlying cause for many individuals’ lack of attention to preventive action, and thus, poor results in achieving target therapeutic values.

### 4.3. Estimating CV Risk Using Various Calculators

The European Society of Cardiology (ESC) prevention guidelines use the 10-year CV mortality risk predictor SCORE as a tool in primary prevention [11]. The SCORE system has been recalibrated in Poland to reflect national mortality and risk factors levels [14], thus, we used the Pol-SCORE system to assess the 10-year risk of fatal CV. While using the Pol-SCORE risk chart, clinicians can identify any individuals with the elevated CV risk (≥1%) of 10-year CVD mortality. Additionally, we used the FRS system to predict the 10-year risk of developing a first fatal or non-fatal CVD event. Recently, guidelines have begun recommending the use of a lifetime risk-assessment in connection with a short-term estimation. Therefore, we also used a newly developed LIFEtime-perspective model for individualizing CardioVascular Disease prevention strategies in apparently healthy people (LIFE-CVD) [21]. Using the LIFE-CVD, there was no significant difference between the moderate and high CV risk participants regarding the estimated lifetime risk of MI, stroke of CV death. This suggests that these participants have a similar unfavourable cardiometabolic and body composition profile resulting in a similar lifetime CV risk. It might be advisable to consider issuing common the guidelines for lifestyle modification and preventive measures for moderate and high-risk classes. It would make it easier for physicians to estimate CV risk, and may imply more aggressive pharmacological and non-pharmacological management of CV risk factors in the moderate risk population, which is clearly high-risk class “in the making”.

Comparing the results of the Pol-SCORE and the 10-year CV risk LIFE-CVD analyses, which estimate both the risk of death due to CVD and the first incident of MI or stroke, the risk of fatal CV risk, estimated using the Pol-SCORE, seems to yield similar results in the low and moderate CV risk groups, and overestimate in the high and very-high CV risk groups. However, we cannot be sure which results reflect real risk, as our study did not have prospective observation.

### 4.4. Limitation and Advantages

This study is limited by a sample from one region, which is an urban environment that may not be representative of the general European population, and the relatively low (41.7%) participation rate, which could have affected the representativeness of the sample. The LIFE-CVD model uses parental history of MI prior to age 60. In our data we obtain information about parental history of MI prior to age 50. Despite the limitations, there are some advantages of the study. We used the data collected from a random sample of local citizens. The variables were collected using standardized questionnaires and methods. The CV risk classes have been carefully evaluated according to the latest recommendations: 2019 ESC/EAS guidelines for the management of dyslipidaemias: Lipid modification to reduce cardiovascular risk [11].

## 5. Conclusions

The participants belonging to the moderate and high CV risk classes have very similar, unfavourable cardiometabolic profiles, which may result in a similar lifetime CV risk of the members of both classes. This may imply the need for more aggressive pharmacological and non-pharmacological management of CV risk factors in the moderate CV risk population. Given the raging epidemics of obesity and diabetes, it may be worth considering parameters reflecting the insulin resistance like abdominal fat distribution to the cardiovascular risk assessment. Given the changes in frequency of risk factors and CV mortality, it would be advisable to carefully validate a CV risk calculators in new prospective studies.

## Figures and Tables

**Figure 1 jcm-10-01584-f001:**
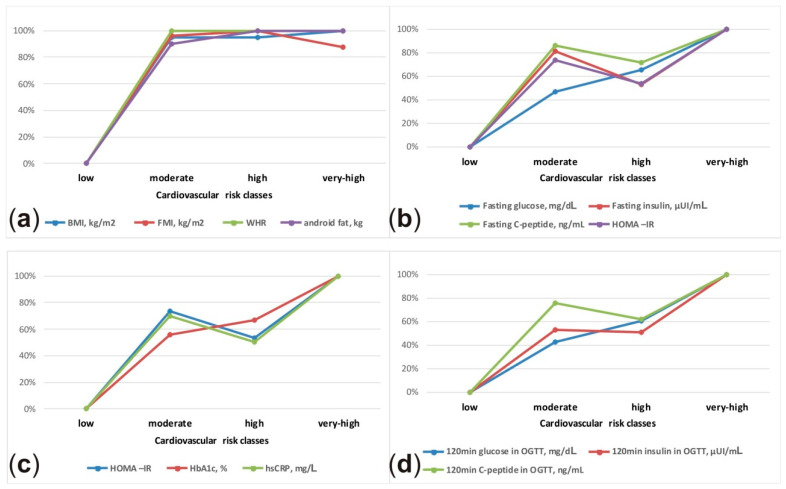
Schematic presentation of the relative differences in anthropometric (**a**), body composition (**a**) and laboratory parameters (**b**–**d**) between cardiovascular risk classes. The lowest value was assumed as 0% and the highest value as 100%.

**Table 1 jcm-10-01584-t001:** Characteristics of the total population and of subgroup according cardiovascular risk and comparisons variables between subgroups (general information, laboratory tests, echocardiography).

Variable	Total Population *n* = 931	Cardiovascular Risk Class			
		Low *n* = 429 (46.1%)	Moderate *n* = 212 (22.8%)	High *n* = 126 (13.5%)	Very High *n* = 164 (17.6%)
Age, years	49.1 ± 15.5	35.7 ± 8.4 ^abc^	54.9 ± 7.7 ^ade^	61.1 ± 10.1 ^bdf^	67.7 ± 7.5 ^cef^
Male sex, *n*	402 (43.2)	155 (36.1) ^c^	**90 (42.5) ^e^**	**56 (44.4) ^f^**	101 (61.6) ^cef^
BPs, mmHg	124.4 ± 17.7	116.2 ± 14.2 ^abc^	**125.9 ± 15.0 ^ae^**	**130.6 ± 16.5 ^bf^**	139.1 ± 18.0 ^cef^
BPd, mmHg	81.7 ± 10.1	78.7 ± 9.1 ^abc^	**84.0 ± 9.6 ^a^**	**84.0 ± 11.1 ^b^**	85.0 ± 10.1 ^c^
BP ≥ 140 and/or ≥90 mmHg	253 (27.2)	57 (13.3) ^abc^	**63 (29.7) ^ae^**	**49 (38.9) ^b^**	84 (51.2) ^ce^
HR, bpm	72.3 ± 10.9	73.9 ± 10.4 ^abc^	**71.0 ± 10.8 ^a^**	**70.9 ± 10.7 ^b^**	70.8 ± 12.0 ^c^
*Laboratory tests*					
NT-proBNP, pg/mL	89.4 ± 190.3	52.9 ± 47.4 ^abc^	**74.6 ± 72.6 ^ae^**	**98.8 ± 96.6 ^bf^**	192.0 ± 408.8 ^cef^
hs-TnT, pg/mL	7.4 ± 5.1	5.4 ± 1.9 ^abc^	7.2 ± 4.2 ^ade^	8.5 ± 6.1 ^bdf^	10.5 ± 7.2 ^cef^
Fasting glucose, mg/dL	102.1 ± 21.0	94.4 ± 9.4 ^abc^	**104.1 ± 21.4 ^ae^**	**107.9 ± 23.9 ^bf^**	115.1 ± 29.9 ^cef^
OGTT 120 min glucose, mg/dL	124.3 ± 39.7	110.0 ± 27.8 ^abc^	**129.1 ± 35.9 ^ae^**	**137.2 ± 46.7 ^bf^**	154.7 ± 49.9 ^cef^
Fasting insulin, µUI/mL	12.3 ± 7.6	10.9 ± 6.8 ^abc^	**13.5 ± 8.0 ^a^**	**12.6 ± 6.2 ^b^**	14.1 ± 9.3 ^c^
OGTT 120 min Insulin, µUI/mL	64.8 ± 64.3	52.4 ± 50.2 ^abc^	**72.2 ± 58.6 ^a^**	**71.5 ± 61.5 ^b^**	89.9 ± 100.2 ^c^
HbA1 c, %	5.5 ± 0.7	5.2 ± 0.42 ^abc^	**5.7 ± 0.6 ^ae^**	**5.8 ± 0.7 ^bf^**	6.1 ± 0.9 ^cef^
HOMA –IR	3.2 ± 2.8	2.6 ± 1.9 ^abc^	**3.7 ± 4.0 ^a^**	**3.4 ± 2.2 ^b^**	4.1 ± 3.2 ^c^
Fasting C-peptide, ng/mL	2.6 ± 1.1	2.2 ± 1.0 ^abc^	**2.8 ± 1.1 ^a^**	**2.7 ± 1.0 ^b^**	2.9 ± 1.3 ^c^
OGTT 120 min C-peptide, ng/mL	8.8 ± 3.8	7.7 ± 3.1 ^abc^	**9.9 ± 3.7 ^a^**	**9.5 ± 3.9 ^b^**	10.6 ± 4.7 ^c^
TC, mg/dL	192.5 ± 40.8	181.0 ± 32.8 ^abc^	199.3 ± 31.1 ^ad^	214.9 ± 48.7 ^bdf^	196.7 ± 53.0^cf^
LDL-C, mg/dL	124.4 ± 37.8	113.9 ± 30.0 ^abc^	**130.8 ± 28.6 ^a^**	**143.5 ± 47.6 ^bf^**	129.1 ± 48.5 ^cf^
HDL-C, mg/dL	62.6 ± 17.3	64.1 ± 16.1 ^c^	**62.5 ± 19.6**	**61.8 ± 16.1**	59.2 ± 18.0 ^c^
TG, mg/dL	113.2 ± 77.6	96.6 ± 82.4 ^abc^	**122.9 ± 70.1 ^a^**	**137.0 ± 75.2 ^b^**	125.7 ± 66.1 ^c^
hsCRP, mg/l	1.7 ± 4.2	1.3 ± 3.2 ^abc^	**2.0 ± 5.0 ^a^**	**1.8 ± 3.7 ^b^**	2.3 ± 5.3 ^c^
Creatinine, μmol/L	70.9 ± 14.9	69.5 ± 14.6 ^bc^	69.0 ± 15.0 ^de^	73.1 ± 14.2 ^bd^	75.0 ± 15.3 ^ce^
CrCl, mL/min	115.0 ± 40.7	126.6 ± 42.8 ^abc^	116.3 ± 40.2 ^ade^	102.2 ± 33.3 ^bd^	92.7 ± 26.8 ^ce^
*Echocardiography*					
LVEF Biplane, %	58.5 ± 5.7	59.6 ± 5.3 ^bc^	**58.7 ± 5.2 ^e^**	**57.8 ± 5.5 ^b^**	55.9 ± 6.9 ^ce^
LVMI, g/m^2^	77.4 ± 20.5	68.5 ± 16.5 ^abc^	**80.9 ± 18.2 ^ae^**	**83.6 ± 19.6 ^bf^**	92.0 ± 22.4 ^cef^
LVMI, ≥95 g/m^2^ women, ≥115 g/m^2^ men	84 (9.3)	9 (2.1)^abc^	**25 (12.1)^a^**	**17 (14.3)^b^**	33 (21.3)^c^
LAVI, mL/m^2^	22.6 ± 7.0	20.6 ± 5.7 ^abc^	**23.5 ± 7.0 ^a^**	**24.0 ± 6.7 ^b^**	25.6 ± 8.6 ^c^
LAVI, >34 mL/m^2^ *	53 (6.2)	6 (1.5) ^abc^	**15 (7.7) ^a^**	**11 (9.6) ^b^**	21 (14.2) ^c^
Diastolic dysfunction *	105 (11.4)	20 (4.7) ^abc^	**22 (10.5) ^ae^**	**19 (15.2) ^b^**	44 (27.7) ^ce^

The data is shown as *n* (%), mean ± SD. BP: blood pressure; BPd: diastolic blood pressure; BPs: systolic blood pressure; bpm: beats per min; CrCl: creatinine clearance using Cockcroft-Gault Equation; HbA1 c: hemoglobin A1 c; HDL-C: high-density lipoprotein; HOMA-IR: homeostasis model assessment of insulin resistance; HR: heart rate; hs-CRP; high- sensitivity C-reactive protein; hs-TnT: high-sensitivity troponin T; kg: kilogram; LAVI: left atrial volume index; LDL-C: low-density lipoprotein; mmHg, millimeters of mercury; LVEF Biplane: left ventricular ejection fraction biplane Simpson’s method; LVMI: left ventricular mass index; m^2^: square meter; NT-proBNP: *n*-terminal pro-brain natriuretic peptide; OGTT: oral glucose tolerance test; SD: standard deviation; TC: total cholesterol; TG: triglycerides; Comparisons variables between subgroups, the same letters in each row (a: between low and moderate CV risk classes; b: between low and high CV risk classes; c: between low and very-high CV risk classes; d: between moderate and high CV risk classes; e: between moderate and very-high CV risk classes; f: between high and very-high CV risk classes) represent significant differences at *p* < 0.05. * Diastolic dysfunction of left ventricle was assessed based on the latest recommendations [13]. No significant differences between the moderate and high risk groups are shown in bold.

**Table 2 jcm-10-01584-t002:** Characteristics of the total population and of subgroup according cardiovascular risk and comparisons variables between subgroups (anthropometric measurements, body composition analysis, subjective well-being).

Variable	Total Population *n* = 931	Cardiovascular Risk Class			
		Low *n* = 429 (46.1%)	Moderate *n* = 212 (22.8%)	High *n* = 126 (13.5%)	Very high *n* = 164 (17.6%)
*Anthropometric measurements and body composition analysis*					
BMI, kg/m^2^	26.8 ± 5.0	24.8 ± 4.4 ^abc^	**28.4 ± 5.0 ^a^**	**28.5 ± 4.2 ^b^**	28.7 ± 4.6 ^c^
BMI < 25 kg/m^2^	330 (35.4)	224 (52.2)^abc^	**49 (23.1) ^a^**	**24 (19.0) ^b^**	33 (20.1) ^c^
BMI 25–29.99 kg/m^2^	352 (37.8)	146 (34.0)	**91 (42.9)**	**52 (41.3)**	63 (38.4)
BMI ≥ 30 kg/m^2^	249 (26.7)	59 (13.8)^abc^	**72 (34.0) ^a^**	**50 (39.7) ^b^**	68 (41.5) ^c^
Body mass, kg	77.2 ± 16.2	73.6 ± 16.2 ^abc^	**80.3 ± 16.1 ^a^**	**80.8 ± 14.7 ^b^**	80.1 ± 15.2 ^c^
Height, cm	169.6 ± 9.9	171.8 ± 9.6 ^abc^	**168.2 ± 9.7 ^a^**	**168.2 ± 10.7 ^b^**	166.8 ± 9.2 ^c^
Waist, cm	87.0 ± 13.5	80.5 ± 11.7 ^abc^	**90.9 ± 12.1 ^af^**	**92.6 ± 11.6 ^b^**	94.8 ± 13.0 ^cf^
Hips, cm	99.52 ± 9.6	97.0 ± 9.2 ^abc^	**101.8 ± 10.2 ^a^**	**102.1 ± 8.3 ^b^**	101.0 ± 9.2 ^c^
Waist, >80 cm women, >94 cm men	636 (68.8)	233 (54.8) ^abc^	**166 (78.7) ^a^**	**106 (84.8) ^b^**	131 (79.9) ^c^
Thigh, cm	58.2 ± 5.9	58.4 ± 6.3 ^c^	**59.2 ± 5.7 ^e^**	**58.5 ± 5.3 ^f^**	56.5 ± 5.1 ^cef^
WHR	0.9 ± 0.1	0.8 ± 0.1 ^abc^	**0.9 ± 0.1 ^ae^**	**0.9 ± 0.1 ^bf^**	0.9 ± 0.1 ^cef^
WHR, ≥0.85 women, ≥0.9 men	464 (50.1%)	134 (31.5) ^abc^	**121 (57.3) ^ae^**	**82 (65.1) ^b^**	127 (77.4) ^ae^
FMI (kg/m^2^)	9.2 ± 3.5	8.0 ± 3.1 ^abc^	**10.3 ± 3.7 ^a^**	**10.4 ± 3.4 ^b^**	10.1 ± 3.3 ^c^
Total fat mass, kg	26.1 ± 9.2	23.1 ± 8.7 ^abc^	**28.8 ± 9.6 ^a^**	**29.1 ± 8.2 ^b^**	27.9 ± 8.4 ^c^
Total lean mass, kg	48.8 ± 10.6	48.2 ± 11.2	**49.0 ± 9.7**	**49.5 ± 10.9**	49.9 ± 9.7
Android fat mass, kg	2.4 ± 1.2	1.9 ± 1.1 ^acb^	**2.8 ± 1.2 ^a^**	**2.9 ± 1.1 ^b^**	2.9 ± 1.2 ^c^
Gynoid fat mass, kg	4.1 ± 1.4	3.9 ± 1.4 ^ab^	**4.3 ± 1.6 ^a^**	**4.3 ± 1.3 ^b^**	3.9 ± 1.2
Gynoid lean mass, kg	7.2 ± 1.6	7.2 ± 1.7	**7.2 ± 1.5**	**7.3 ± 1.6**	7.4 ± 1.5
Legs fat mass, kg	7.7 ± 2.8	7.6 ± 2.8	**8.2 ± 3.0 ^e^**	**7.9 ± 2.6 ^f^**	7.1 ± 2.4 ^ef^
Legs lean mass, kg	16.9 ± 4.0	1.7 ± 4.2	**1.7 ± 3.8**	**1.7 ± 4.0**	1.7 ± 3.8
Visceral mass, kg	1.2 ± 1.0	0.7 ± 0.7 ^abc^	**1.4 ± 0.9 ^ae^**	**1.7 ± 1.0 ^b^**	1.9 ± 1.1 ^ce^
A/G fat ratio	0.6 ± 0.2	0.5 ± 0.2 ^abc^	**0.6 ± 0.2 ^ae^**	**0.7 ± 0.2 ^bf^**	0.8 ± 0.2 ^cef^
*Subjective well-being*					
SWLS	23.07 ± 5.30	23.6 ± 5.3	**22.7 ± 5.0**	**22.8 ± 5.3**	22.4 ±5.6
EQ-VAS	76.7 ± 14.7	80.9 ± 13.4 ^abc^	**76.3 ± 13.6 ^ae^**	**72.6 ± 15.1 ^b^**	68.9 ^ce^
BDI	7.02 ± 6.58	6.2 ± 6.3 ^bc^	**6.6 ± 5.7 ^e^**	**8.0 ± 6.8 ^b^**	8.9 ± 7.7 ^ce^

The data is shown as *n* (%), mean ± SD. A: android; BDI: Beck Depression Inventory; BMI: body mass index; EQ-VA: Euro Quality of Life Visual Analogue Scale; FMI: fat mass index; G: gynoid; SD: standard deviation; SWLS: Satisfaction With Life Scale; WHR: waist-hip ratio; Comparisons variables between subgroups, the same letters in each row (a: between low and moderate CV risk classes; b: between low and high CV risk classes; c: between low and very-high CV risk classes; d: between moderate and high CV risk classes; e: between moderate and very-high CV risk classes; f: between high and very-high CV risk classes) represent significant differences at *p* < 0.05. No significant differences between the moderate and high risk groups are shown in bold.

**Table 3 jcm-10-01584-t003:** Medical history of the total population and of subgroup according cardiovascular risk and comparisons variables between subgroups.

Medical history	Total Population *n* = 931	Cardiovascular Risk Class			
		Low *n* = 429 (46.1%)	Moderate *n* = 212 (22.8%)	High *n* = 126 (13.5%)	Very high *n* = 164 (17.6%)
History of hypertension	275 (29.6)	32 (7.5) ^abc^	**77 (36.8) ^ae^**	**60 (47.6) ^bf^**	106 (64.6) ^cef^
Well-controlled BP in patients diagnosed with hypertension *	78 (28.4)	11 (34.4)	**23 (29.9)**	**18 (30.0)**	26 (24.5)
Undiagnosed hypertension	107 (11.5)	44 (10.3) ^a^	24 (11.3) ^ad^	21 (16.7) ^df^	18 (11.0) ^f^
History of hypercholesterolemia	290 (31.1)	58 (13.5) ^abc^	**98 (46.7) ^a^**	**57 (45.2) ^b^**	77 (47.0)^c^
Well-controlled lipid profile in patients with diagnosed hypercholesterolemia **	39 (13.4)	11 (19.0) ^c^	22 (22.4) ^de^	3 (5.3) ^d^	3 (3.9) ^ce^
Undiagnosed hypercholesterolemia ***	399 (42.9)	149 (34.7) ^abc^	**99 (46.7) ^a^**	**66 (52.4) ^b^**	85 (51.8) ^c^
History of diabetes	71 (7.6)	2 (0.5) ^abc^	**18 (8.5) ^ae^**	**14 (11.1) ^b^**	37 (22.7) ^ae^
Well controlled glucose in patients diagnosed with diabetes ****	51 (71.8)	1 (50)	**16 (88.9)**	**9 (64.3)**	25 (67.6)
Undiagnosed diabetes *****	57 (6.1)	5 (1.2)	**16 (7.5)**	**10 (7.9)**	26 (15.9)
History of atrial fibrillation	27 (2.9)	1 (0.2) ^abc^	**6 (2.9) ^a^**	**9 (7.2) ^b^**	11 (6.8) ^c^
Currently smoking	186 (20.1)	86 (20.1)	**39 (18.4)**	**23 (18.5)**	38 (23.5)

The data are shown as *n* (%). BP: blood pressure; BPs: systolic blood pressure; BPd: diastolic blood pressure; CV: cardiovascular; HbA1c: hemoglobin A1c; HDL: high-density lipoprotein; LDL: low-density lipoprotein; OGTT: oral glucose tolerance test; TC: total cholesterol; TG: triglycerides; * BPs < 130 and BPd < 80 mmHg below 65 years old, BPs < 140 and BPd < 80 mmHg 65–80 years old, BPs < 150 and BPd < 80 mmHg over 80 years old. ** LDL-c < 116 mg% in low CV class, <100 mg% in moderate CV class, <70 mg% in high CV class, <55 mg% in very-high CV class *** TC > 190 mg% or LDL-c > 116 mg% in low CV class, > 100 mg% in moderate CV class, >70 mg% in high CV class, >55 mg% in very-high CV class **** HbA1c < 7.0% ***** Fasting glucose ≥ 126 mg/dL or OGGT 120 min. glucose ≥ 200 mg/dL. Comparisons variables between subgroups, the same letters in each row (a: between low and moderate CV risk classes; b: between low and high CV risk classes; c: between low and very-high CV risk classes; d: between moderate and high CV risk classes; e: between moderate and very-high CV risk classes; f: between high and very-high CV risk classes) represent significant differences at *p* < 0.05. No significant differences between the moderate and high risk groups are shown in bold.

**Table 4 jcm-10-01584-t004:** The value of different predicting tools of the total population and of subgroup according cardiovascular risk and comparisons variables between subgroups.

CV Risk Calculators	Total Population	Cardiovascular Risk Class			
		Low	Moderate	High	Very High
Pol-SCORE, %	4.0 ± 4.9	0.5 ± 0.3 ^abc^	2.5 ± 1.1 ^ade^	6.1 ± 2.3 ^bdf^	15.3 ± 6.0 ^cef^
FRS-Lipids, %	8.6 ± 8.2	2.4 ± 1.7 ^abc^	8.6 ± 4.7 ^ade^	13.3 ± 6.0 ^bdf^	23.9 ± 7.1 ^cef^
FRS-BMI, %	10.9 ± 9.6	3.2 ± 2.2 ^abc^	11.9 ± 6.1 ^ade^	17.7 ± 8.3 ^bdf^	27.0 ± 5.3 ^cef^
LIFE-CVD 10-year risk, %	4.9 ± 3.9	1.3 ± 0.7 ^abc^	3.2 ± 1.4 ^ade^	5.0 ± 1.7 ^bdf^	10.1 ± 4.8 ^cef^
LIFE-CVD Lifetime risk, %	17.3 ± 8.4	11.5 ± 3.4 ^abc^	**16.4 ± 6.7 ^ae^**	**17.8 ± 8.4 ^bf^**	22.4 ± 10.2 ^cef^

The data are shown as mean ± SD. BMI: body mass index; CV: cardiovascular; CVD: cardiovascular disease; FRS: Framingham Risk Score; LIFE-CVD: LIFEtime-perspective model for individualizing CardioVascular Disease prevention strategies in apparently healthy people; SCORE: Systematic Coronary Risk Estimation; SC: standard deviation. Comparisons variables between subgroups, the same letters in each row (a: between low and moderate CV risk classes; b: between low and high CV risk classes; c: between low and very-high CV risk classes; d: between moderate and high CV risk classes; e: between moderate and very-high CV risk classes; f: between high and very-high CV risk classes) represent significant differences at *p* < 0.05. No significant differences between the moderate and high risk groups are shown in bold.

## Data Availability

The data set we generated during and/or analyzed during the current study are not publicly available due to confidentiality issues but are available from the corresponding author on request.

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
