# Peer review of "A Similar Lifetime CV Risk and a Similar Cardiometabolic Profile in the Moderate and High Cardiovascular Risk Populations: A Population-Based Study"

_jcm, 2021, doi:10.3390/jcm10081584_

Round 1

Reviewer 1 Report

Comments to the Authors:

This is cross-sectional study which recruited local general population using mayor’s office database . A total of 931 individuals aged 20-79 were included in this study. Anthropometric measurement, questionnaire collecting demographic data and CV risk factors, blood test and echocardiography were conducted. The study population was divided into CV ris classes according to “Cardiovascular Risk Class”, which was created based on the combination of 2019 ESC/EAS Guideline and Pol-SCORE (scoring system based on age, gender, smoking, BPs and total cholesterol).   

The indices of muscle mass at various sites did not differ across the risk groups. The indices of well-being was not associated with risk groups. The higher risk class was associated with higher various other risk scores including FRS etc..  However, metabolic profiles and LIFE-CVD Lifetime risk were similar between moderate and high risk classes. From these findings the Authors concluded that more aggressive management for moderate risk population is needed.

Major comments:

First of all, the significance of this study remains obscure. It looks like Pol-SCORE is calculated based on the classic risk factors. The Reviewer does not agree to the conclusion reached by the Authors, and just felt that this classic scoring system might be insufficient to discriminate the future CV risk of moderate and high risk groups.  

Too much data are presented in the Tables. It is difficult for the readers to focus on the important findings that the Authors want to emphasize.

The Reviewer cannot find Figure 1.

Minor comments:

Table 1. Please specify the definition of “diastolic dysfunction”.

Table 1. Cockcroft-Gault provides estimation of creatinine clearance, not GFR.

Table 1. Footnote. WHR is not listed in the Table.

Author Response

Responses for the Reviewer

  1. First of all, the significance of this study remains obscure. It looks like Pol-SCORE is calculated based on the classic risk factors. The Reviewer does not agree to the conclusion reached by the Authors, and just felt that this classic scoring system might be insufficient to discriminate the future CV risk of moderate and high risk groups.  

            The European Society of Cardiology (ESC) recommends the use of the Systematic COronary Risk Evaluation - SCORE [1] enables the assessment of the risk of death from cardiovascular causes within 10 years on the basis of information on age, sex, systolic blood pressure, total cholesterol concentration and smoking [3]. These are known and validated risk factors. The ESC recommends, whenever possible, the use of a SCORE scale validated for a given country. It was recalibrated in Poland [2], thus we use Pol-SCORE system to assess the 10-year risk of fatal CV for individuals aged 40-70. The validation in 2015 of Pol-SCORE tables for the Polish population included an analysis of two main elements: epidemiological data on the prevalence of smoking, as well as mean age, systolic blood pressure and total cholesterol in 5-year age groups; these data were obtained from the NATPOL 2011 study, and data on deaths from cardiovascular causes in 2011 in 5-year age groups (codes I10-I99 according to ICD).  This is the currently valid method of assessing cardiovascular risk in Poland.

Considering the change in the characteristics of the population due to the increasing number of overweight and obese individuals, and therefore insulin resistance, we suggests that it may be worth considering parameters reflecting the insulin resistance like abdominal fat distribution to the cardiovascular risk assessment.

  1. Mach, F.; Baigent, C.; Catapano, A.L.; Koskinas, K.C.; Casula, M.; Badimon, L.; Chapman, M.J.; De Backer, G.G.; Delgado, V.; Ference, B.A., et al. 2019 ESC/EAS Guidelines for the management of dyslipidaemias: lipid modification to reduce cardiovascular risk. Eur Heart J 2020, 41, 111-188, doi:10.1093/eurheartj/ehz455
  2. Zdrojewski, T.; Jankowski, P.; Bandosz, P.; Bartus, S.; Chwojnicki, K.; Drygas, W.; Gaciong, Z.; Hoffman, P.; Kalarus, Z.; Kazmierczak, J., et al. [A new version of cardiovascular risk assessment system and risk charts calibrated for Polish population]. Kardiol Pol 2015, 73, 958-961, doi:10.5603/KP.2015.0182.
  3. Too much data are presented in the Tables. It is difficult for the readers to focus on the important findings that the Authors want to emphasize.

No significant differences between the moderate and high cardiovascular risk groups are shown in bold to highlight important data.

  1. The Reviewer cannot find Figure 1.

We apologize for the inconvenience, the figure was attached separately - has been supplemented.

Figure 1. Schematic presentation of the relative differences in anthropometric, body composition and laboratory parameters between cardiovascular risk classes. The lowest value was assumed as 0% and the highest value as 100%.

  1. Table 1. Please specify the definition of “diastolic dysfunction”.

It has been supplemented in Table 1: “* Diastolic dysfunction of left ventricle was assessed based on the latest recommendations [13]”.

  1. Table 1. Cockcroft-Gault provides estimation of creatinine clearance, not GFR.

It has been corrected in Table 1: “CrCl: creatinine clearance using Cockcroft-Gault Equation”

  1. Table 1. Footnote. WHR is not listed in the Table.

“WHR: waist-hip ratio” has been removed from the footnotes, this table does not contain these data.

All changes to the manuscript are marked in red.

Thank you for your review, it will improve the quality of the article.

Kind regards,

Karol Kaminski

Reviewer 2 Report

It is with great interest that I have read the manuscript entitled "A similar lifetime CV risk and a similar cardiometabolic proflie in the moderate and high cardiovascular risk populations: a population-based study" from Chlabicz and colleagues.

The authors aimed to investigate the cardiovascular (CV) profiles for patients with different CV risks in Poland, in order to improve CV prevention, management and outcomes.

Although this concept is timely, and reduction of CV risk and deaths is of great importance I do have some major concerns prior to publication of this manuscript.

Major concerns:

The authors fail to formulate a clear conclusion and message. The discussion section is too long, has a very descriptive nature and does not put the authors finding into perspective.  The authors must improve this section, and discuss their findings (e.g. the discrepancy between 'subjective well being analysis' and higher CV risk, or the fact that estimated lifetime CV risk is no different between moderate and high CV risk) and discuss potential clinical consequences of their CV risk strategy in further detail. 

I like the idea of specific CV risk profiles - perhaps the authors can elaborate on these profiles a bit more, both in results (e.g. central illustration) and discussion section (e.g. how these different CV risk proflies may have clinical relevance and how they may improve prevention/treatment strategy).

Minor concerns:

The tables are hard to read and statistical testing remains unclear. It would be good to display overall significance, and add an extra box for p-values. Statistical testing between subgroups can then be addes for those variables with overall p-value <0.05.  

Methods: Study population, line 68-78: personally, I would like to see these results in the "results section". 

There are some differences in font size; please correct this.

Author Response

Responses for the Reviewer  

  1. The authors fail to formulate a clear conclusion and message. The discussion section is too long, has a very descriptive nature and does not put the authors finding into perspective.  The authors must improve this section, and discuss their findings (e.g. the discrepancy between 'subjective well being analysis' and higher CV risk, or the fact that estimated lifetime CV risk is no different between moderate and high CV risk) and discuss potential clinical consequences of their CV risk strategy in further detail. 

Cardiovascular diseases (CVD) is still the leading cause of death and disability in the European population. In our clinical work we notice a reluctance to take preventive actions in the moderate cardiovascular risk class, not only among patients but also in the medical community. We know from clinical experience that these people consider themselves healthy. They disregard recommendations and these actions are even supported by some doctors. The main aim of this study was to investigate whether people with moderate cardiovascular risk, who often consider themselves healthy are clinically closer to the high-risk class than the low risk one. This is what we actually found and this finding should be strongly emphasized. Now we changed the introduction of the manuscript to better reflect this idea.

To our knowledge, this is the first such study in a population with such a detailed and accurate breakdown into cardiovascular risk classes. Therefore, we quite broadly refer to studies from other countries as well as discuss risk factors prevalence and management. We also emphasize that timely acknowledgement of inceased cardiovascular risk may allow initiation of appropriate treatment that improves survival and reduces disability.This is especially important in the moderate CV risk category, where these people do not know and are reluctant to know about their diseases.

In order to asses subjective well-being we have conducted an additional analysis (Supplementary material, Table S1), and we found that life satisfaction was not related to the actual risk factors such as the presence of hypertension, hypercholesterolaemia or hyperglycemia, while a strong unfavorable relationship was shown with the diagnosis of hypertension, hypercholesterolaemia or hyperglycemia itself.  Therefore, we have made our conclusions that participants do not want to know about diseases that affect CV risk, because they could lower their life satisfaction. This may be an underlying cause for many individuals’ lack of attention to preventive action, and so, poor results in achieving target therapeutic values.

Therefore, in our opinion, it is necessary to highlight this phenomenon and educate the community in this regard.

We add Supplementary materials:

Table S1 Results of SWLS, EQ-VAS, BDI analysis in general population

Variable

SWLS

P

EQ-VAS

P

BDI

P

History of hypertension

no

23.4±5.3

0.019

79.1±13.7

<0.001

6.5±6.3

<0.001

yes

22.4±5.2

70.8±15.9

8.4±7.1

History of hypercholesterolemia

no

23.4±5.2

0.041

77.9±14.9

<0.001

6.6±6.5

<0.001

yes

22.5±5.5

73.8±14.4

8.0±6.7

History of diabetes

no

23.1±5.3

0.215

77.4±14.5

<0.001

6.8±6.5

<0.001

yes

22.4±4.8

66.9±15.9

10.21±7.2

BP  ≥140 and/or ≥90mmHg

no

23.1±5.4

0.674

77.1±14.8

0.047

7.1±6.5

0.484

yes

22.9±5.0

75.3±14.9

6.9±6.8

Hypercholesterolemia*

no

23.2±5.4

0.717

80.1±13.5

<0.001

7.0±6.3

0.924

yes

23.0±5.3

75.3±15.1

7.0±6.7

Hyperglycemia**

no

23.2±5.3

0.939

78.0±14.3

<0.001

6.7±6.5

0.002

yes

23.2±5.2

71.4±13.9

8.6±6.5

BDI: Beck Depression Inventory; BP: blood pressure; CV: cardiovascular; EQ-VA: Euro Quality of Life Visual Analogue Scale; LDL-C: low-density cholesterol; OGTT: oral glucose tolerance test; SWLS: Satisfaction With Life Scale

*Total cholesterol >190mg% or LDL-C  >116mg% in low CV class, >100mg% in moderate CV class, >70mg% in high CV class, >55mg% in very-high CV class;

** Fasting glucose ≥ 126 mg/dl or OGTT 120min. ≥ 200mg/dl

  1. I like the idea of specific CV risk profiles - perhaps the authors can elaborate on these profiles a bit more, both in results (e.g. central illustration) and discussion section (e.g. how these different CV risk proflies may have clinical relevance and how they may improve prevention/treatment strategy).

 The figure was attached separately - has been supplemented.

Figure 1. Schematic presentation of the relative differences in anthropometric, body composition and laboratory parameters between cardiovascular risk classes. The lowest value was assumed as 0% and the highest value as 100%.

  1. The tables are hard to read and statistical testing remains unclear. It would be good to display overall significance, and add an extra box for p-values. Statistical testing between subgroups can then be addes for those variables with overall p-value <0.05.  

There is information in the footnotes of the tables: „Comparisons variables between subgroups, the same letters in each row represent significant differences at ˂ 0.05”.

No significant differences between the moderate and high cardiovascular risk groups are shown in bold to highlight important data.

  1. Methods: Study population, line 68-78: personally, I would like to see these results in the "results section". 

             The “Study Population” section was moved to the “Results” section.

  1. There are some differences in font size; please correct this.

             Font has been improved.

All changes to the manuscript are marked in red.

Thank you for your review, it will improve the quality of the article.

Kind regards,

Karol Kaminski

Round 2

Reviewer 1 Report

The quality of the paper a little bit improved, although the Reviewer still feels the presentation of the results is redundant and can be improved. 

Reviewer 2 Report

I would like to thank the authors for their changes in this manuscript. It has significantly improved quality and readability of this manuscript and I have no further comments.